# EXPLAIN LIKE I'M FIVE: USING LLMS TO IMPROVE PDE SURROGATE MODELS WITH TEXT

## ABSTRACT

Solving Partial Differential Equations (PDEs) is ubiquitous in science and engineering. Computational complexity and difficulty in writing numerical solvers has motivated the development of machine learning techniques to generate solutions quickly. Many existing methods are purely data driven, relying solely on numerical solution fields, rather than known system information such as boundary conditions and governing equations. However, the recent rise in popularity of Large Language Models (LLMs) has enabled easy integration of text in multimodal machine learning models. In this work, we use pretrained LLMs to integrate various amounts known system information into PDE learning. Our multimodal approach significantly outperforms our baseline model, FactFormer, in both next-step prediction and autoregressive rollout performance on the 2D Heat, Burgers, Navier-Stokes, and Shallow Water equations. Further analyis shows that pretrained LLMs provide highly structured latent space that is consistent with the amound of system information provided through text.

## 1 INTRODUCTION

Solving Partial Differential Equations (PDEs) is the cornerstone of many areas of science and engineering, from quantum mechanics to fluid dynamics. While traditional numerical solvers often have rigorous error bounds, they are limited in scope, where different methods are required for different governing equations, and different regimes even for a single governing equation. In the area of fluid dynamics, especially, solvers that are designed for Navier Stokes equations generally will not perform optimally in both the laminar and turbulent flow regimes.

Recently, machine learning methods have exploded in popularity to address these downsides in traditional numerical solvers. The primary aim is to reduce time-to-solution and bypass expensive calculations. Physics Informed Neural Networks (PINNs)Raissi et al. (2019) have been widely successful in small-scale systems. More recently, neural operators(Li et al., 2021; Lu et al., 2021; Li et al., 2023a) have improved upon PINNs, showing promise for larger scale, general purpose surrogate models. However, these models generally are purely data-driven, and do not use any additional, known, system information.

Additionally, owing to the success of large language models (LLMs) in many other domains, such as robotics(Kapoor et al., 2024; Bartsch & Farimani, 2024), design(Kumar et al., 2023; Badagabettu et al., 2024; Jadhav & Farimani, 2024; Jadhav et al., 2024), some works have begun incorporating LLMs into the PDE surrogate model training pipeline. Universal Physics Solver (UPS)(Shen et al., 2024) incorporates pretrained LLMs, but uses limited text descriptions that do not fully utilize LLM capabilities. Unisolver(Zhou et al., 2024) takes in a LaTeX description of the system as a prompt, but uses MLP encoders for additional system information such as boundary conditions, which does not fully explore the capabilities of the LLM. ICON-LM(Yang et al., 2024) uses longer text descriptions, but trains the LLM to make numerical predictions from input data and captions, adding additional complexity to the model architecture. Additionally, by using softmax-based attention for numerical predictions, benchmarks are limited to 1D.

**Contributions:** In this work, we aim to more fully utilize LLM understanding in PDE surrogate modeling and incorporate text information into neural operators. To that end, we introduce a novel multimodal PDE framework given in figure 1. This framework is built on top of a FactFormer(Li

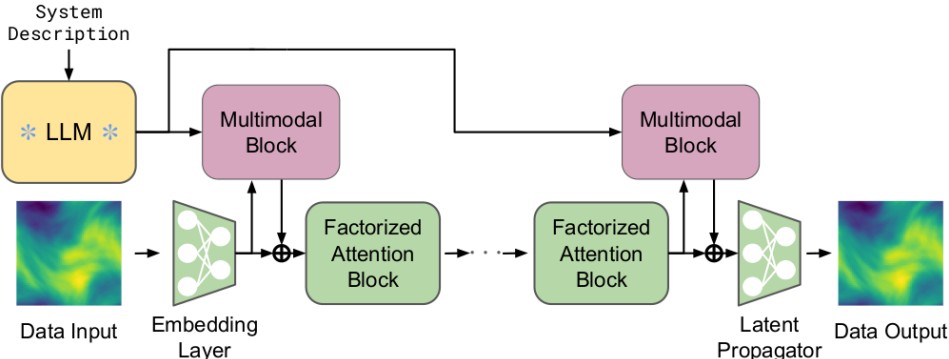

Figure 1: Multimodal FactFormer adds system information through a multimodal block.

et al., 2023b) and successfully incorporates system information through text descriptions. The cross-attention based multimodal block is given in figure 2. We benchmark this framework on the Heat, Burgers, Navier-Stokes, and Shallow Water equations to provide a wide variety of physical behavior. Various boundary conditions, initial conditions, and operator coefficients are used, making these benchmarks more challenging than existing data sets. We test various levels of system information, and incorporate this as conditioning information into FactFormer.

## 2 RELATED WORK

**Neural Solvers:** Neural solvers have been developed largely to counter the drawbacks of traditional numerical methods. Physics Informed Neural Networks (PINNs)(Raissi et al., 2019) incorporate governing equations through a soft constraint on the loss function. PINNs have been shown to be effective in small-scale problems, but tend to be difficult to train(Wang et al., 2022; Rathore et al., 2024). Neural operators(Kovachki et al., 2023) were developed and show improvement over PINNs on a large variety of PDE learning tasks. Based on the universal operator approximation theorem, neural operators learn a functional that maps input functions to solution functions. Neural operators such as Fourier Neural Operator (FNO)(Li et al., 2021), based on kernel integral blocks in Fourier space, DeepONetLu et al. (2021), based on parameterizing the input functions and an embedding of the points at which the functions are evaluated, and OFormer(Li et al., 2023a), based on softmax-free attention. Despite very promising results, neural operators are usually purely data-driven and do not use any system information outside of solution fields.

**Utilizing System Information** While these neural operators perform well, they are often purely data driven. More recent works have incorporated various different aspects of the governing systems. Takamoto et al. (2023a) incorporates operator coefficients(Takamoto et al., 2023a) through the CAPE module. While coefficients are important in determining system behavior, system parameters such as boundary conditions, forcing terms, and geometry can play an equally important role, and cannot be incorporated through CAPE. Further work incorporates governing equations into models. Lorsung et al. (2024) developed the PITT framework that uses a transformer-based architecture to process system information as text. Additionally, PROSE(Liu et al., 2023) poses a multi-objective task to both make a prediction and complete partially correct governing equation. However, these frameworks rely on notational consistency between samples, and do not offer an easy integration of different geometries. Lastly, Hao et al. (2023) introduced novel Heterogeneous Normalized Attention and Geometric Gating mechanisms for flexible GNOT model. GNOT is able to incorporate different system information, such as coefficients and geometry. This flexible architecture can incorporate many different system parameters, but requires additional implementation details for each additional modality, and may not be able to capture qualitative aspects of systems, such as flow regime.

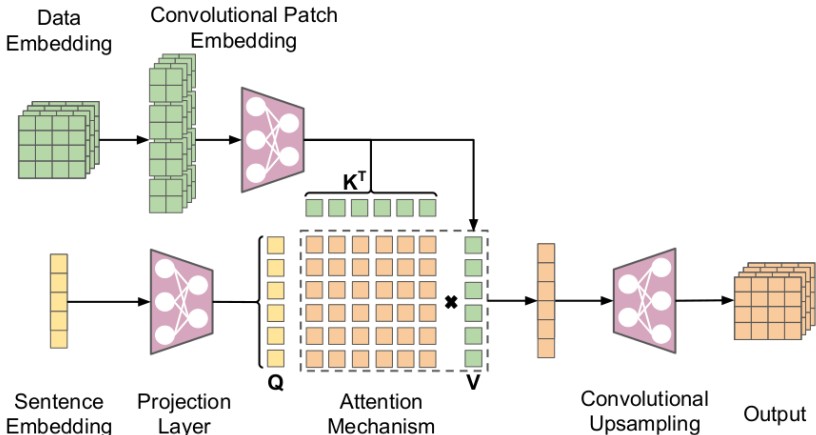

Figure 2: Cross-Attention is used to integrate sentence embeddings with data embeddings.

## 3 DATA GENERATION

We generate data from the Heat, Burgers, and Incompressible Navier Stokes equations, which are popular benchmarks for fluid dynamics surrogate models. Additionally, we benchmark on the Shallow Water equations from PDEBench(Takamoto et al., 2023b). Data for the Heat and Burgers equations are generated using Py-PDE(Zwicker, 2020) due to ease of simulating different boundary conditions, and data is generated for Navier Stokes using code from Fourier Neural Operator(Li et al., 2021). A diverse data set is generated in order to create a more challenging setup than existing benchmarks, that often have the same operator coefficients and boundary conditions for all samples, only varying the initial conditions. In our case, we use different initial conditions, operator coefficients, and boundary conditions. While existing data sets offer distinct challenges with regards to governing equations, they do not offer a lot of data diversity with regard to system parameters, which LLMs are well-suited to handle. PDEBenchTakamoto et al. (2023b), for example, has multiple 2D data sets that represent a variety of physical processes. However, only eight different operator coefficient combinations are used for the 2D Compressible Navier Stokes equations, all with the same boundary conditions. In order to fully utilize the capabilities of pretrained LLMs, as well as present a more challenging benchmark, we vary boundary conditions and operator coefficients for the Heat and Burgers equations in 2D, and vary viscosity and forcing term amplitude for the Incompressible Navier-Stokes equations in 2D.

### 3.1 HEAT EQUATION

The heat equation models a diffusive process and is given below in equation 1, where we are predicting the temperature distribution at each time step.

$$u_t = \beta \nabla^2 u \tag{1}$$

We generate data on a simulation cell given by: $\Omega = [-0.5, 0.5]^2$ on a 64x64 grid with boundary conditions sampled from $\partial \Omega \in \{Neumann, Dirichlet, Periodic\}$. The values $v$ for the Neumann and Dirichlet boundary conditions are sampled uniformly: $v \in \mathcal{U}(-0.1, 0.1)$. All four walls have the same boundary condition type and value for a given simulation. Initial conditions are chosen from three different distributions: exponential field given by $f(x, y) = \exp\left(100(x + y)^2\right)$, sum of sine and cosine given by $f(x, y) = \sin(c_1 \pi x) + \cos(c_2 \pi x)$, and product of sines given by $f(x, y) = \sin(c_1 \pi x) \sin(c_2 \pi y)$ for $c_1, c_2 \in \{2, 4, 6, 8\}$. Lastly, our diffusion coefficient was sampled randomly according to $\beta \in \mathcal{U}(0.005/\pi, 0.02/\pi)$. We simulated each trajectory for 2 seconds on a 64x64 grid. These distribution of diffusion coefficients was chosen so the diffusive dynamics behaved on approximately the same scale as the advection dynamics of Burgers Equation given below.

## 3.2 BURGERS' EQUATION

Burgers equation models shock formation in fluid waves, given below in equation 2, where we are predicting the [height], $u$ at each time step.

$$u_t = \beta \nabla u - \alpha u \cdot \nabla^2 u \tag{2}$$

System specifications for the Heat equation are identical to Heat equation, given above, with the exception of the advection term. Our advection coefficient is sampled according to $\alpha_x, \alpha_y \in \mathcal{U}(-0.5, 0.5)$.

## 3.3 INCOMPRESSIBLE NAVIER-STOKES EQUATIONS

Third, we generate data from the Incompressible Navier-Stokes equations in vorticity form, given below in equation 3.

$$\partial_t w(x,t) + u(x,t) \cdot \nabla w(x,t) = \nu \Delta w(x,t) + f(x)$$
$$\nabla \cdot u(x,t) = 0 \tag{3}$$
$$w(x,0) = w_0(x)$$

The Navier-Stokes data generated here follows the setup from Lorsung et al. (2024). Our simulation cell is given by: $\Omega = [0,1]^2$ with periodic boundary conditions. Our viscosity is sampled according to $\nu \in \{10^{-9}, 2 \cdot 10^{-9}, 3 \cdot 10^{-9}, \ldots, 10^{-8}, 2 \cdot 10^{-8}, \ldots, 10^{-5}\}$, and our forcing term amplitude is sampled according to $A \in \{0.001, 0.002, 0.003, \ldots, 0.01\}$. The initial condition is given by a Gaussian random field. Data generation is done on a 256x256 grid that is evenly downsampled to a 64x64 grid for this work.

## 3.4 SHALLOW-WATER EQUATIONS

Lastly, we take the Shallow-Water data set from PDEBench(Takamoto et al., 2023b). In this case, we are predicting the water height $h$ at each time.

$$\partial_t h + \partial_x hu + \partial_y hv = 0$$
$$\partial_t hu + \partial_x \left( u^2 h + \frac{1}{2} g_r h^2 \right) = -g_r h \partial_x b \tag{4}$$
$$\partial_t hv + \partial_y \left( v^2 h + \frac{1}{2} g_r h^2 \right) = -g_r h \partial_y b$$

Our simulation cell is: $\Omega = [-2.5, 2.5]^2$ on a 64x64 grid, with Neumann boundary conditions, with 0 gradient on the boundary.

For each combination of initial conditions and boundary conditions, 900 samples were generated with randomly generated coefficients. For Navier Stokes, we did the vorticity formulation, from FNO, and varied viscosity and forcing term amplitude, with random gaussian field initial condition. 30 second simulation. Also used PDEBench's shallow water data set. These are all single-channel. In our experiments, B refers to incorporating boundary condition information through text, C refers to incorporating coefficient information through text, and Q refers to incorporating qualitative information through text.

## 4 METHODS

The multimodal appraoch developed here uses full sentence descriptions of systems from our data sets, given in section 4.1. The cross-attention based multimodal block is uses both FactFormer embeddings as well as LLM embeddings and is described in section 4.2.

## 4.1 SYSTEM DESCRIPTIONS

To explore how well our LLMs are able to incorporate different amounts of text information, we describe each of the Heat, Burgers, and Navier-Stokes equations with varying levels of detail. We

will build a complete sentence description here, with all possible sentence combinations given in appendix A. At a base level, we can describe the basic properties of each governing equation, as well as identifying the equation. For Burgers equation, we generally have stronger advection forces, so we describe it as:

```
Burgers equation models a conservative system that can develop shock wave
        discontinuities.  Burgers equation is a first order quasilinear
                    hyperbolic partial differential equation.
```

Next, boundary condition information is added, in this case Neumann boundary conditions:

```
This system has Neumann boundary conditions.  Neumann boundary conditions
have a constant gradient.  In this case we have a gradient of ∂u_neumann on
                             the boundary.
```

Third, we can add operator coefficient information:

```
In this case, the advection term has a coefficient of α_x in the x
   direction, α_y in the y direction, and the diffusion term has a
                        coefficient of β.
```

Lastly, we can add qualitative information. The aim of this is to capture details about the system that are intuitive to practitioners, but difficult to encode mathematically. In this case, we have an advection dominated system:

```
This system is advection dominated and does not behave similarly to heat
            equation.  The predicted state should develop shocks.
```

Our complete sentence description is passed into a pretrained LLM that is used to generate embeddings. These embeddings are then used as conditioning information for our model output.

## 4.2 MULTIMODAL PDE LEARNING

The backbone of our multimodal surrogate model is the FactFormer(Li et al., 2023b) and our framework is given in figure 1. FactFormer was chosen because it provides a fast and accurate benchmark model. We add our system description as conditioning information both before and after factorized attention blocks. FactFormer is a neural operator that learns a functional $\mathcal{G}_\theta$ that maps from input function space $\mathbb{A}$ to solution function space $\mathbb{U}$ as $\mathcal{G}_\theta : \mathbb{A} \to \mathbb{U}$, with parameters $\theta$. In our case, we are specifically learning the functional conditioned on our system information $s$. For a given input function $a$ evaluated at points $x$, we are leaning the the operator given in equation 5. That is, our network learns to make predictions for our solution function $u$ also evaluated at points $x$.

$$G(u)(\boldsymbol{x}) = \mathcal{G}_\theta(a)(\boldsymbol{x}, \boldsymbol{s}) \tag{5}$$

Sentences are first passed through an LLM to generate an embedding. The embedding is combined with the data through cross-attention, seen in figure 2, where our sentence embedding is the queries and the data embedding is the keys and values. The conditioned embedding is then added back to our data embedding, given below in equation 6. Our data embedding is embedding with convolutional patch embedding, mapping our data to a lower dimension as $f_{data} : \mathbb{R}^{b \times h \times w \times h_{FF}} \to \mathbb{R}^{b \times p \times h_{FF}}$, where $b$ is our batch size, $h_{FF}$ is our FactFormer embedding dimension, and $p$ is the number of patches, which is defined by kernel size and convolutional stride. Our sentence embeddings are projected to higher dimensional space to match our data embedding dimension, $f_{sentence} : \mathbb{R}^{b \times h_{LLM}} \to \mathbb{R}^{b \times p \times 1} \to \mathbb{R}^{b \times p \times h_{FF}}$ through two successive MLPs, where $h_{LLM}$ is the LLM output dimension. Once we have the data embedding $\boldsymbol{z}_{data} = f_{data}(\boldsymbol{a})$ and our sentence embedding $\boldsymbol{z}_{sentence} = f_{sentence}(\boldsymbol{s})$, for data sample $\boldsymbol{a}$ and sentence description $\boldsymbol{s}$, we calculate the multimodal embedding $\boldsymbol{z}_{multimodal}$ using multihead attention(Vaswani et al., 2017) given below in equation 6.

$$\boldsymbol{z}_{multimodal} = \text{Concat}\left(\text{head}_1, \dots, \text{head}_h\right) \boldsymbol{W}^O$$

$$\text{and head}_i = \text{softmax}\left(\frac{\boldsymbol{z}_{sentence}\boldsymbol{W}_i^Q\left(\boldsymbol{z}_{data}\boldsymbol{W}_i^K\right)^T}{\sqrt{d_k}}\right)\boldsymbol{z}_{data}\boldsymbol{W}_i^V \tag{6}$$

$z_{multimodal}$ is then returned from our multimodal block and used in the FactFormer architecture. We use cross-attention here instead of self-attention so that our sentence embeddings can be used to learn useful context for our data embeddings. After cross-attention, our multimodal embedding is upsampled back to the data embedding dimension using deconvolution(Zeiler et al., 2010) $f_{deconv}$ : $\mathbb{R}^{b \times p \times h_{FF}} \to \mathbb{R}^{b \times h \times w \times h_{FF}}$ given in equation 7:

$$z_{data} = z_{data} + f_{deconv}\left(z_{multimodal}\right) \tag{7}$$

This multimodal block is agnostic to both the data-driven backbone as well as the LLM. We compare `Llama 3.1 8B`(AI@Meta, 2024) and `all-mpnet-base-v2` from the Sentence-Transformer package(Reimers & Gurevych, 2019) as our pretrained LLMs. `Llama 3.1 8B` was used because of its good performance across a wide variety of benchmarks, as well as its small size allowing us to use only a single GPU to generate sentence embeddings. The word embeddings are averaged for each sentence to provide a single embedding that can be used in our projection layer. `all-mpnet-base-v2` was used because it is designed to generate embeddings from sentences that are useful for tasks like sentence similarity. In all of our experiments, the LLM is frozen and a projection head is trained. This significantly improves training time by allowing us to generate the sentence embeddings before training, and avoiding expensive gradient computations for our LLMs.

## 5 RESULTS

We benchmark our multimodal model against its baseline variant on a number of challenging tasks. Our data vary the distribution of initial conditions, operator coefficients, and boundary conditions, which provides a much more challenging setting than many existing benchmarks. In each experiment, the combined data set is the Heat, Burgers, and Navier-Stokes data sets, where Shallow Water is used solely for fine-tuning, with a 90-10-10 train-validation-test split. 4,000 samples per equation are used for Heat, Burgers, and Navier-Stokes data unless otherwise noted, and we use the entire 1,000 sample Shallow-Water data set when using more samples for our other data sets. We use one frame of data to predict the next frame, which allows us to asses how well our multimodal approach captures known system information, which is the operator $G_\theta\left(a\right)\left(\boldsymbol{x}, \boldsymbol{s}\right)_{t=n} \to u(\boldsymbol{x})_{t=n+1}$. That is the input function at points $x$ and time $t = n$ conditioned on sentence information $s$ is used to predict our solution function at time $t = n+1$. Using only a single step of input here presents the additional challenge that it is difficult to infer boundary conditions and impossible to infer operator coefficients from only a single frame of data. Our temporal horizon is limited to 21 steps for each equation during training (initial condition and 20 steps), and 40 steps during autoregressive rollout evaluation. Relative $L^2$ Error(Li et al., 2021) is used for both training and next-step evaluation in all of our experiments. Autoregressive rollout error is reported as Mean Squared Error. In our results, we used SentenceTransformer (ST)(Reimers & Gurevych, 2019) and Llama 3.1 8B (Llama)(AI@Meta, 2024), using various combinations of boundary condition (B), coefficient (C) and qualitative (Q) information. Each reported result is the mean and standard deviation across three random seeds.

### 5.1 NEXT-STEP PREDICTION

First, we evaluate next-step predictive accuracy, where we take one frame of data and use it to predict the next across our entire temporal window. We see in figure 3 that our multimodal FactFormer significantly outperforms baseline FactFormer when using SentenceTransformer for our Heat, Burgers, and Shallow Water data sets, both with and without transfer learning, where we have small improvement on our Navier-Stokes benchmark when doing standard training. Our SentenceTransformer FactFormer had an average of 21.4% less error across all of our data sets for standard training and average of 20.6% less error than baseline FactFormer when using transfer learning. Additionally, we see smaller improvement over baseline results when using Llama as our LLM, with some notable increases in error on the Shallow Water and Navier Stokes data sets.

### 5.2 AUTOREGRESSIVE ROLLOUT

Second, we perform autoregressive rollout starting from our initial condition for 40 steps. This evaluates how well our model is able to temporally extrapolate beyond our training horizon. In this case, we use the models trained in section 5.1 with no additional fine-tuning. Accumulated

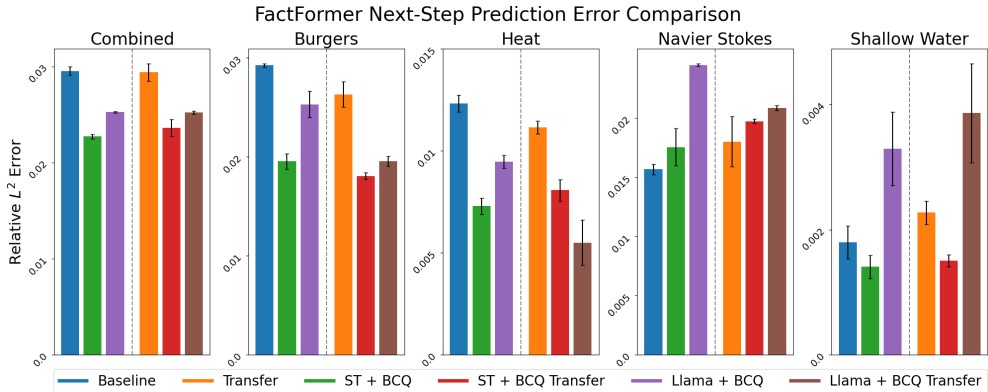

Figure 3: Comparison of next-step prediction relative $L^2$ error for baseline FactFormer, FactFormer + ST, and Factformer + Llama with and without transfer learning.

rollout error is presented in table 1. We see that our SentenceTransformer FactFormer has lowest accumulated error for four of five data sets both for standard training and transfer learning, and outperforms baseline across all data sets. Our Llama FactFormer also outperforms baseline across all of our data sets except Shallow Water with standard training. For SentenceTransformer Factformer, we have an average reduction in accumulated error of 64.2% and 84.7% across all of our data set for standard training and transfer learning, respectively. For Llama FactFormer, we have an average reduction in accumulated error of 6.2% and 81.2% for standard and transfer learning, respectively. When removing Shallow Water for Llama Factformer with standard training, we have an average reduction in error of 58.0%.

We note that accumulated autoregressive rollout error for our Shallow Water benchmark is much higher for our transfer learning experiments than the standard training. This instability been noted before in Lorsung et al. (2024) when training errors are very low. In our case, training error for the Shallow Water data set is approximately an order of magnitude lower than other data sets. Error plots are given in figure 8 in Appendix C.

Table 1: Comparison of accumulated mean squared error ($\times 10^2$) for baseline FactFormer, Fact-Former + ST, and baseline + Llama with and without transfer learning.

|  | Baseline | ST | Llama | Transfer | ST + Transfer | Llama + Transfer |
|---|---|---|---|---|---|---|
| Combined | $1.36 \pm 0.03$ | $\mathbf{0.78 \pm 0.04}$ | $0.90 \pm 0.02$ | $3.89 \pm 0.01$ | $\mathbf{0.80 \pm 0.09}$ | $0.86 \pm 0.04$ |
| Burgers | $3.49 \pm 0.15$ | $\mathbf{0.97 \pm 0.10}$ | $1.41 \pm 0.42$ | $12.17 \pm 0.13$ | $\mathbf{0.85 \pm 0.07}$ | $1.03 \pm 0.10$ |
| Heat | $1.08 \pm 0.10$ | $\mathbf{0.16 \pm 0.01}$ | $0.47 \pm 0.09$ | $4.61 \pm 0.03$ | $0.30 \pm 0.17$ | $\mathbf{0.13 \pm 0.07}$ |
| Navier Stokes | $1.52 \pm 0.46$ | $0.54 \pm 0.25$ | $\mathbf{0.26 \pm 0.09}$ | $2.56 \pm 0.03$ | $\mathbf{0.25 \pm 0.00}$ | $0.40 \pm 0.08$ |
| Shallow Water | $0.12 \pm 0.09$ | $\mathbf{0.05 \pm 0.04}$ | $0.35 \pm 0.36$ | $156.60 \pm 1.64$ | $\mathbf{51.22 \pm 60.16}$ | $65.70 \pm 21.50$ |

## 5.3 DATA SCALING

Using the same training setup as in section 5.1, we benchmark our multimodal models against baseline as we increase the amount of data for the Heat and Burgers data sets, given in figure 4, with scaling for the remaining datasets given in Appendix E. For the Heat and Burgers data sets, we see that our multimodal approach consistently outperforms baseline for our both large and small data settings, both with SentenceTransformer and Llama, and for both standard training and transfer learning. We add more data the relative improvement increases due to the plateau in our base models.

## 5.4 ABLATION STUDY

Lastly, we perform an ablation study based on text information, to determine which components of our text description are most helpful during the learning process. We compare our baseline model against our multimodal model with SentenceTransformer using 4,000 samples for each equation.

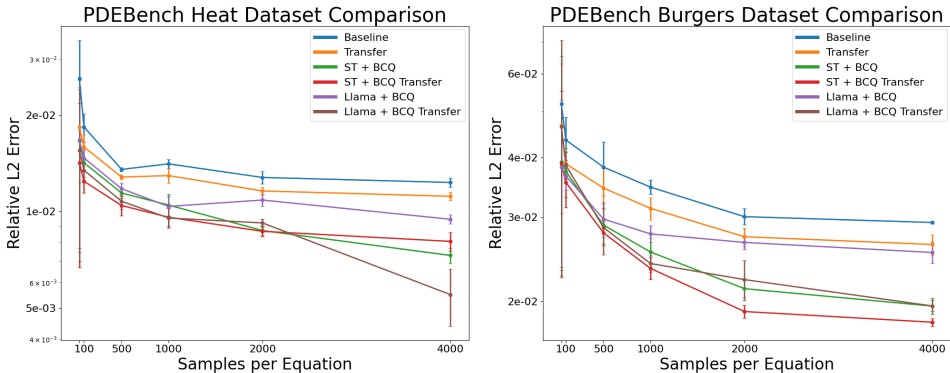

Figure 4: Comparison of next-step prediction relative $L^2$ error for baseline FactFormer, FactFormer + ST, and Factformer + Llama with and without transfer learning as the amount of data increases.

Both baseline and multimodal models were trained with transfer learning. Results for next-step prediction are given in figure 5, and autoregressive accumulated rollout errors are given in 2. In both experiments, we see a correlation between richness of sentence description and performance, but that every level of text information improves upon baseline model with transfer learning. Additional error plots and results are given in appendix C.

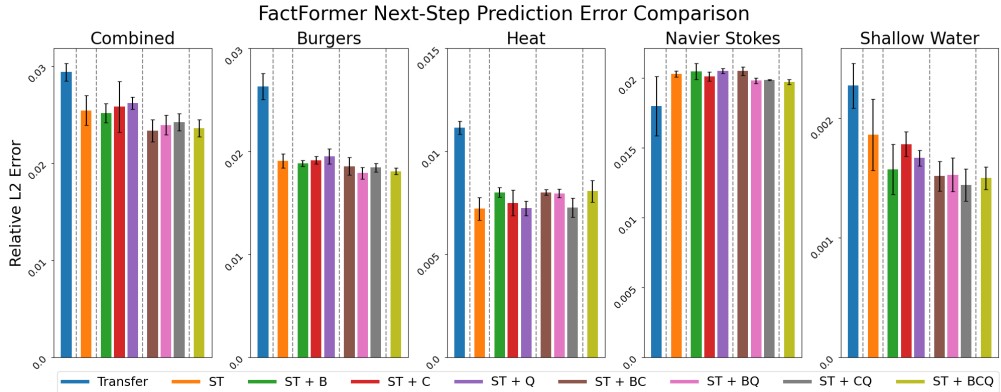

Figure 5: Comparison of next-step prediction relative $L^2$ error for baseline FactFormer and Fact-Former + ST with varying levels of sentence description, trained with transfer learning.

Table 2: Comparison of accumulated mean squared error $(\times 10^2)$ for baseline FactFormer and Fact-Former + ST with varying levels of sentence description, trained with transfer learning.

| | Transfer | None | B | C | Q | BC | BQ | CQ | BCQ |
|---|---|---|---|---|---|---|---|---|---|
| Combined | $3.89 \pm 0.01$ | $0.84 \pm 0.02$ | $0.85 \pm 0.01$ | $0.85 \pm 0.08$ | $0.87 \pm 0.01$ | $0.82 \pm 0.07$ | $\mathbf{0.80 \pm 0.02}$ | $0.82 \pm 0.07$ | $0.80 \pm 0.09$ |
| Burgers | $12.17 \pm 0.13$ | $1.02 \pm 0.04$ | $0.86 \pm 0.08$ | $1.10 \pm 0.05$ | $1.04 \pm 0.07$ | $0.86 \pm 0.04$ | $0.86 \pm 0.09$ | $1.00 \pm 0.07$ | $\mathbf{0.85 \pm 0.07}$ |
| Heat | $4.61 \pm 0.03$ | $0.37 \pm 0.02$ | $0.30 \pm 0.19$ | $0.38 \pm 0.03$ | $0.38 \pm 0.05$ | $0.20 \pm 0.04$ | $\mathbf{0.19 \pm 0.03}$ | $0.38 \pm 0.02$ | $0.30 \pm 0.17$ |
| Navier Stokes | $2.56 \pm 0.03$ | $\mathbf{0.19 \pm 0.15}$ | $0.29 \pm 0.04$ | $0.26 \pm 0.02$ | $0.33 \pm 0.06$ | $0.27 \pm 0.02$ | $0.27 \pm 0.01$ | $0.24 \pm 0.03$ | $0.25 \pm 0.00$ |
| Shallow Water | $156.60 \pm 1.64$ | $82.06 \pm 33.18$ | $\mathbf{10.02 \pm 9.98}$ | $56.15 \pm 50.17$ | $51.89 \pm 33.97$ | $63.49 \pm 50.75$ | $43.81 \pm 50.04$ | $65.07 \pm 75.05$ | $51.22 \pm 60.16$ |

# 6 DISCUSSION

The aim of our multimodal approach is to incorporate known system information through sentence descriptions, rather than complex data conditioning strategies. We see for our next-step prediction and autoregressive rollout tasks that a multimodal approach is able to significantly improve performance over baseline. SentenceTransformer tends to improve performance more than Llama, and this is likely due to the nature of the embeddings generated by each LLM. SentenceTransformer

is trained to generate useful embeddings at the sentence level, whereas Llama is trained to generate word-level embeddings. In order to utilize Llama for out framework, we need to average over all word-level embeddings, which can lose subtle information in our embeddings. The sentence-level embeddings from SentenceTransformer, on the other hand, allows us to use these embeddings directly from the LLM in our multimodal cross-attention block.

We can visualize the embeddings from SentenceTransformer for each level of text description as well in our combined dataset. In figure 6, we have t-SNE embeddings for sentences that simply identify the equation compared against t-SNE embeddings with our full text description, that is with boundary condition, operator coefficient, and qualitative information. In our combined data set we only have samples from the Heat, Burgers, and Navier-Stokes equations, which is reflected in the three clusters seen. However, with our full text description, we have a significantly richer embedding. We see large clusters with subregions with our Heat and Burgers samples, matching our continuous distributions for operator coefficients and boundary condition values [analyze further?]. This shows how we are able to easily introduce significantly more structure into our data, making the learning task easier. t-SNE plots were generated with Chan et al. (2019).

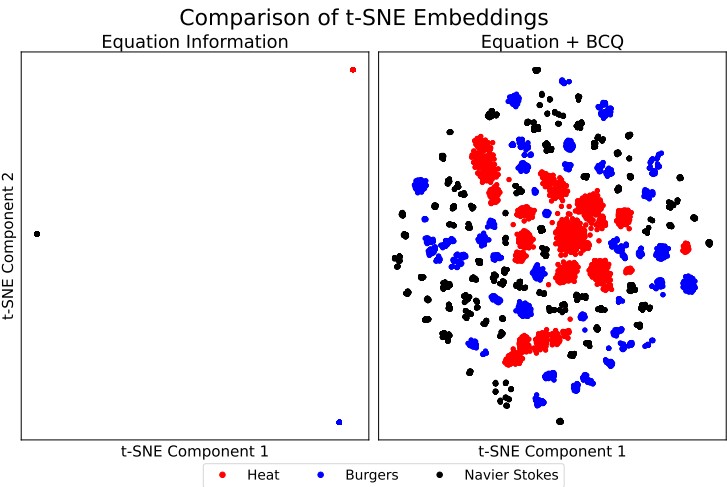

Figure 6: t-SNE embeddings for sentence-level embeddings generated by SentenceTransformer for basic equation description and equation description with boundary conditions, operator coefficients, and qualitative information.

## 7 CONCLUSION

We introduced a multimodal approach to PDE surrogate modeling that utilizes LLMs and sentence descriptions of our systems that significantly improves performance for a variety of tasks. Our multimodal approach allows us to easily incorporate system information that both captures quantitative and qualitative aspects of our governing equations. Analysis of our sentence embeddings shows that we are able to capture increasing amounts of underlying structure in our data by simply adding more information through text, rather than introducing additional mathematical approaches, such as constrained loss functions that are popular with PINNs.

While this direction is promising, performance could be further improved with tuning out LLMs directly, rather than just training an embedding layer that uses LLM output, and we leave this to future work. Additionally, pretraining strategies have shown success in various LLM applications from natural language processing to PDE surrogate modeling. While this does introduce significant computational overhead, it may improve performance and is also a potential future direction to take. Lastly, we only benchmarked the FactFormer, but this framework and multimodal block is not specific to FactFormer, and may prove useful for improving other physics-based models such as FNO and DeepONet.

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

# A   SENTENCE DESCRIPTIONS

Basic information:

Heat:
: The Heat equation models how a quantity such as heat diffuses through a given region. The Heat equation is a linear parabolic partial differential equation.

Burgers:
: Burgers equation models a conservative system that can develop shock wave discontinuities. Burgers equation is a first order quasilinear hyperbolic partial differential equation.

Navier Stokes:
: The incompressible Navier Stokes equations describe the motion of a viscous fluid with constant density. We are predicting the vorticity field, which describes the local spinning motion of the fluid.

Shallow Water:
: The Shallow-Water equations are a set of hyperbolic partial differential equations that describe the flow below a pressure surface in a fluid.

Coefficient Information:

Heat:
: In this case, the diffusion term has a coefficient of $\beta$.

Burgers:
: In this case, the advection term has a coefficient of $\alpha_x$ in the x direction, $\alpha_y$ in the y direction, and the diffusion term has a coefficient of $\beta$.

Navier-Stokes:
: In this case, the viscosity is $\nu$. This system is driven by a forcing term of the form f(x,y) = A*(sin(2*pi*(x+y)) + cos(2*pi*(x+y))) with amplitude A=$A$.

Shallow-Water:
: N/A

Boundary Condition Information:

| Heat: | Periodic | This system has periodic boundary conditions. The simulation space is a torus. |
| | Neumann: | This system has Neumann boundary conditions. Neumann boundary conditions have a constant gradient. In this case we have a gradient of  on the boundary. |
| | Dirichlet: | This system has Dirichlet boundary conditions. Dirichlet boundary conditions have a constant value. In this case we have a value of  on the boundary. |
| Burgers: | Periodic | This system has periodic boundary conditions. The simulation space is a torus. |
| | Neumann: | This system has Neumann boundary conditions. Neumann boundary conditions have a constant gradient. In this case we have a gradient of  on the boundary. |
| | Dirichlet: | This system has Dirichlet boundary conditions. Dirichlet boundary conditions have a constant value. In this case we have a value of  on the boundary. |
| Navier-Stokes: | Periodic: | This system has periodic boundary conditions. The simulation cell is a torus. |
| Shallow-Water: | Neumann | This system has homogeneous Neumann boundary conditions with a derivative of 0 at the boundary. |

Qualitative Information:

| | | |
|---|---|---|
| Heat: | $\beta > 0.01$ | This system is strongly diffusive. The predicted state should look smoother than the inputs. |
| | $\beta \leq 0.01$ | This system is weakly diffusive. The predicted state should looke smoother than the inputs. |
| Burgers: | $\frac{\|\alpha\|_2}{\beta} > 100$ | This system is advection dominated and does not behave similarly to heat equation. The predicted state should develop shocks. |
| | $\frac{\|\alpha\|_2}{\beta} \leq 100$ | This system is diffusion dominated and does behave similarly to heat equation. The predicted state should look smoother than the inputs. |
| Navier-Stokes: | $\nu \geq 1E-6$ | This system has high viscosity and will not develop small scale structure. |
| | $1E-6 > \nu \geq 1E-8$ | This sytem has moderate viscosity and will have some small scale structure. |
| | $1E-8 > \nu$ | This system has low viscosity and will have chaotic evolution with small scale structure. |
| | $A \geq 7E-4$ | This system has a strong forcing term and evolution will be heavily influenced by it. |
| | $7E-4 > A \geq 3E-4$ | This system has a moderate forcing term and evolution will be moderately influenced by it. |
| | $3E-4 > A$ | This system has a weak forcing term and evolvution will be weakly influenced by it. |
| Shallow-Water: | | This system simulates a radial dam break. Waves propagate outward in a circular pattern. |

## B HYPERPARAMETERS

### B.1 ARCHITECTURE

We use identical architecture across all of our experiments, given in table 3. We note that the hidden dimension controls both the FactFormer embedding dimension as well as the LLM projection head embedding dimension. This is done so we can use our multimodal cross-attention block without any additional data projections. Our convolutional patch embedding used a kernel size of 8, and our

Table 3: FactFormer architecture hyperparameters.

| Model | Depth | Hidden Dim | Head Dim | Heads | Kernel Multiplier | Latent Multiplier |
|---|---|---|---|---|---|---|
| FactFormer | 1 | 128 | 64 | 4 | 2 | 2 |

convolutional upsampling layer used a ConvTranspose layer with kernel size of 8 with a stride size of 4, followed by two fully connected layers with widths of 128, and GELU activation. Additionally, our sentence embedding projection layer used four fully-connected layers with size [384/4096, 256, 256, 128, no. patches] to match the number of patches from our patch embedding, with ReLU activation, then upsamples our channel dimension of 1 to match our hidden dimension of 128 using two fully connected layers with width 128 and GELU activation.

### B.2 TRAINING

Training hyperparameters were generally kept as similar as possible, although instability in training was noted for our Multimodal FactFormer at times during finetuning. Generally, lower learning rate and higher weight decay led to more stable training. Table 4 has training hyperparameters for our 4000 samples/equation case. We trained each model for 1000 epochs for each experiment. In each epoch, each trajectory is used, where the specific input/output frame for each trajectory is randomly sampled. Additional tuning was necessary for training stability for FactFormer + Llama with small

Table 4: Training hyperparameters for FactFormer, FactFormer + ST and FactFormer + Llama

| Model | Batch Size | Learning Rate | Weight Decay | Finetune Learning Rate | Finetune Weight Decay |
|---|---|---|---|---|---|
| FactFormer | 64 | 1e-4 | 1e-8 | 1e-4 | 1e-8 |
| FactFormer (Transfer) | 64 | 1e-4 | 1e-8 | 1e-4 | 1e-8 |
| FactFormer + ST | 64 | 1e-4 | 1e-8 | 1e-4 | 1e-8 |
| FactFormer + ST (Transfer) | 64 | 1e-4 | 1e-8 | 1e-4 | 1e-8 |
| FactFormer + Llama | 64 | 5e-4 | 1e-8 | 5e-5 | 1e-8 |
| FactFormer + Llama (Transfer) | 64 | 5e-4 | 1e-8 | 5e-5 | 1e-8 |

amounts of data. Hyperparameters are given in table 5. No additional tuning was necessary for baseline FactFormer of FactFormer + ST.

Table 5: Data scaling hyperparameters for FactFormer + Llama

| Samples/Equation | Batch Size | Learning Rate | Weight Decay | Finetune Learning Rate | Finetune Weight Decay |
|---|---|---|---|---|---|
| 50 | 128 | 5e-4 | 1e-7 | 1e-4 | 1e-7 |
| 100 | 128 | 5e-4 | 1e-7 | 1e-4 | 1e-7 |
| 500 | 128 | 5e-4 | 1e-7 | 1e-4 | 1e-7 |
| 1000 | 128 | 5e-4 | 1e-7 | 1e-4 | 1e-7 |
| 2000 | 256 | 5e-4 | 1e-8 | 5e-5 | 1e-8 |
| 4000 | 64 | 5e-4 | 1e-8 | 5e-5 | 1e-8 |
| 50 (Transfer) | 128 | 5e-4 | 1e-7 | 1e-4 | 1e-7 |
| 100 (Transfer) | 128 | 5e-4 | 1e-7 | 1e-4 | 1e-7 |
| 500 (Transfer) | 128 | 5e-4 | 1e-7 | 1e-4 | 1e-7 |
| 1000 (Transfer) | 128 | 5e-4 | 1e-7 | 1e-4 | 1e-7 |
| 2000 (Transfer) | 256 | 5e-4 | 1e-8 | 5e-5 | 1e-8 |
| 4000 (Transfer) | 64 | 5e-4 | 1e-8 | 5e-5 | 1e-8 |

## C    ADDITIONAL NEXT-STEP AND ROLLOUT RESULTS

Table 6 below gives numerical values for figure 3. Figure 7 plots the values of table 1.

Table 6: Next-Step prediction results

|  | Baseline | ST | Llama | Transfer | ST + Transfer | Llama + Transfer |
|---|---|---|---|---|---|---|
| Combined | $2.95 \pm 0.04$ | $\mathbf{2.27 \pm 0.02}$ | $2.52 \pm 0.01$ | $2.94 \pm 0.09$ | $2.36 \pm 0.09$ | $2.52 \pm 0.02$ |
| Burgers | $2.93 \pm 0.02$ | $\mathbf{1.95 \pm 0.08}$ | $2.53 \pm 0.13$ | $2.63 \pm 0.13$ | $1.81 \pm 0.03$ | $1.96 \pm 0.05$ |
| Heat | $1.23 \pm 0.04$ | $\mathbf{0.73 \pm 0.04}$ | $0.95 \pm 0.03$ | $1.12 \pm 0.03$ | $0.81 \pm 0.05$ | $0.55 \pm 0.11$ |
| Navier Stokes | $\mathbf{1.57 \pm 0.04}$ | $1.75 \pm 0.16$ | $2.45 \pm 0.01$ | $1.80 \pm 0.21$ | $1.97 \pm 0.02$ | $2.08 \pm 0.02$ |
| Shallow Water | $0.18 \pm 0.03$ | $\mathbf{0.14 \pm 0.02}$ | $0.33 \pm 0.06$ | $0.23 \pm 0.02$ | $0.15 \pm 0.01$ | $0.39 \pm 0.08$ |

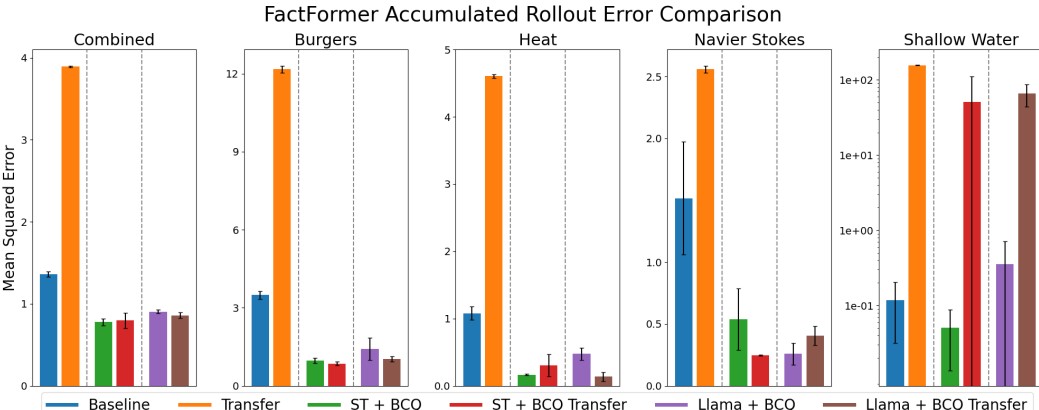

Figure 7: Comparison of accumulated mean squared error ($\times 10^2$) for baseline FactFormer, FactFormer + ST, and baseline + Llama with and without transfer learning.

Figure 8 plots the autoregressive rollout error against timestep. Table 1 presents the sum over timesteps of these plots. Errors are first averaged over each sample for a given random seed, then the mean and standard deviation across seeds is plotted.

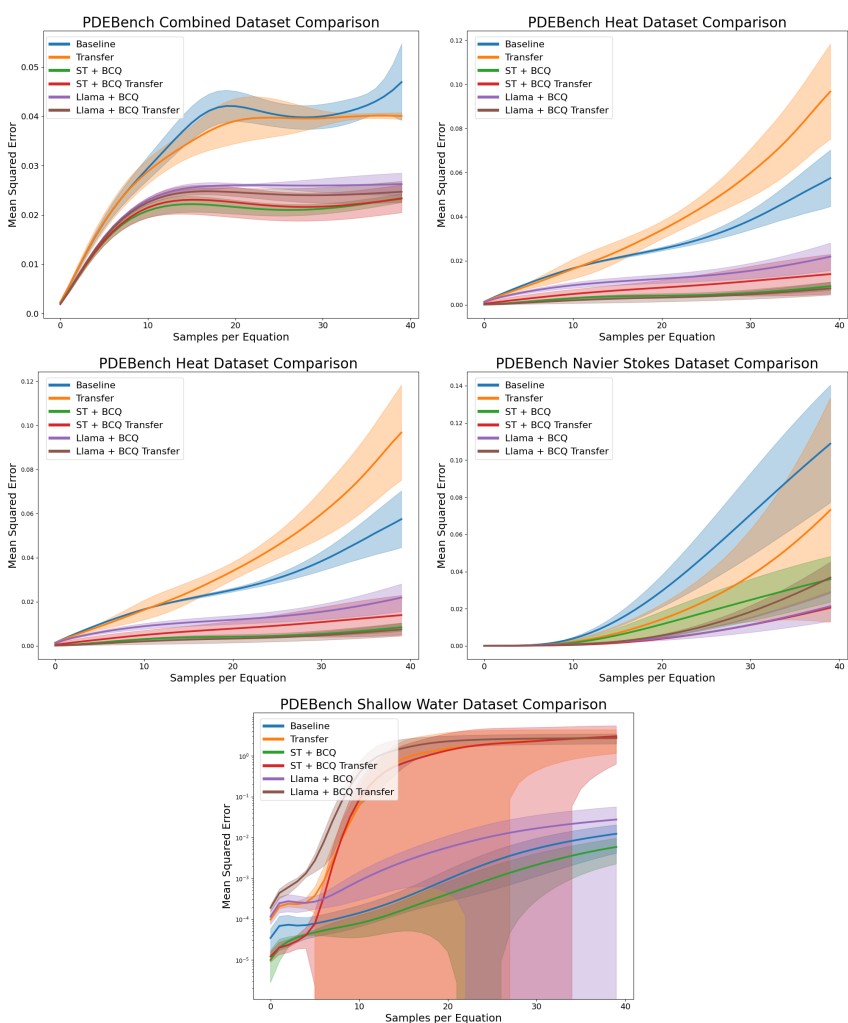

Figure 8: Autoregressive rollout MSE for each dataset.

# D    ABLATION STUDY ADDITIONAL RESULTS

Table 7 below gives numerical values for figure 5. Figure 9 plots the values of table 2.

Table 7: Ablation study next-step prediction error

|  | Baseline | None | B | C | Q | BC | BQ | CQ | BCQ |
|---|---|---|---|---|---|---|---|---|---|
| Combined | $3.22 \pm 0.05$ | $2.86 \pm 0.02$ | $2.93 \pm 0.13$ | $\mathbf{2.81 \pm 0.07}$ | $2.89 \pm 0.09$ | $2.86 \pm 0.06$ | $2.88 \pm 0.06$ | $2.85 \pm 0.01$ | $2.86 \pm 0.04$ |
| Burgers | $3.13 \pm 0.17$ | $2.35 \pm 0.11$ | $2.32 \pm 0.20$ | $2.32 \pm 0.12$ | $2.44 \pm 0.16$ | $\mathbf{2.31 \pm 0.21}$ | $2.35 \pm 0.10$ | $2.33 \pm 0.12$ | $2.34 \pm 0.12$ |
| Heat | $1.30 \pm 0.07$ | $1.00 \pm 0.06$ | $0.97 \pm 0.06$ | $1.00 \pm 0.07$ | $1.02 \pm 0.08$ | $\mathbf{0.95 \pm 0.06}$ | $0.96 \pm 0.05$ | $1.00 \pm 0.06$ | $0.96 \pm 0.06$ |
| Navier Stokes | $\mathbf{2.01 \pm 0.07}$ | $2.20 \pm 0.02$ | $2.20 \pm 0.01$ | $2.19 \pm 0.03$ | $2.23 \pm 0.14$ | $2.20 \pm 0.02$ | $2.12 \pm 0.02$ | $2.11 \pm 0.03$ | $2.12 \pm 0.03$ |
| Shallow Water | $0.23 \pm 0.01$ | $0.16 \pm 0.00$ | $0.16 \pm 0.00$ | $0.17 \pm 0.01$ | $0.17 \pm 0.02$ | $\mathbf{0.15 \pm 0.01}$ | $0.16 \pm 0.01$ | $0.16 \pm 0.01$ | $0.16 \pm 0.01$ |

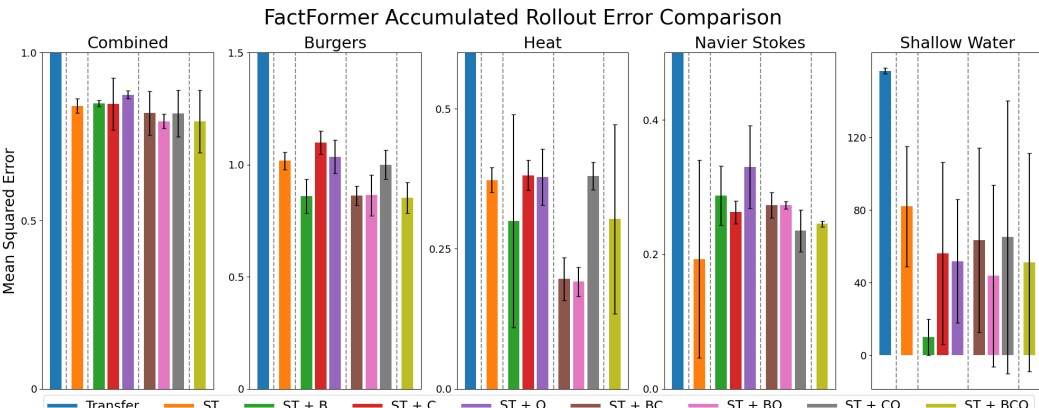

Figure 9: Comparison of next-step prediction relative $L^2$ error for baseline FactFormer and Fact-Former + ST with varying levels of sentence description, trained with transfer learning. The y-axis is truncated to show results from our multimodal FactFormer more clearly.

Figure 10 plots the autoregressive rollout error against timestep. Table 2 presents the sum over timesteps of these plots. Errors are first averaged over each sample for a given random seed, then the mean and standard deviation across seeds is plotted.

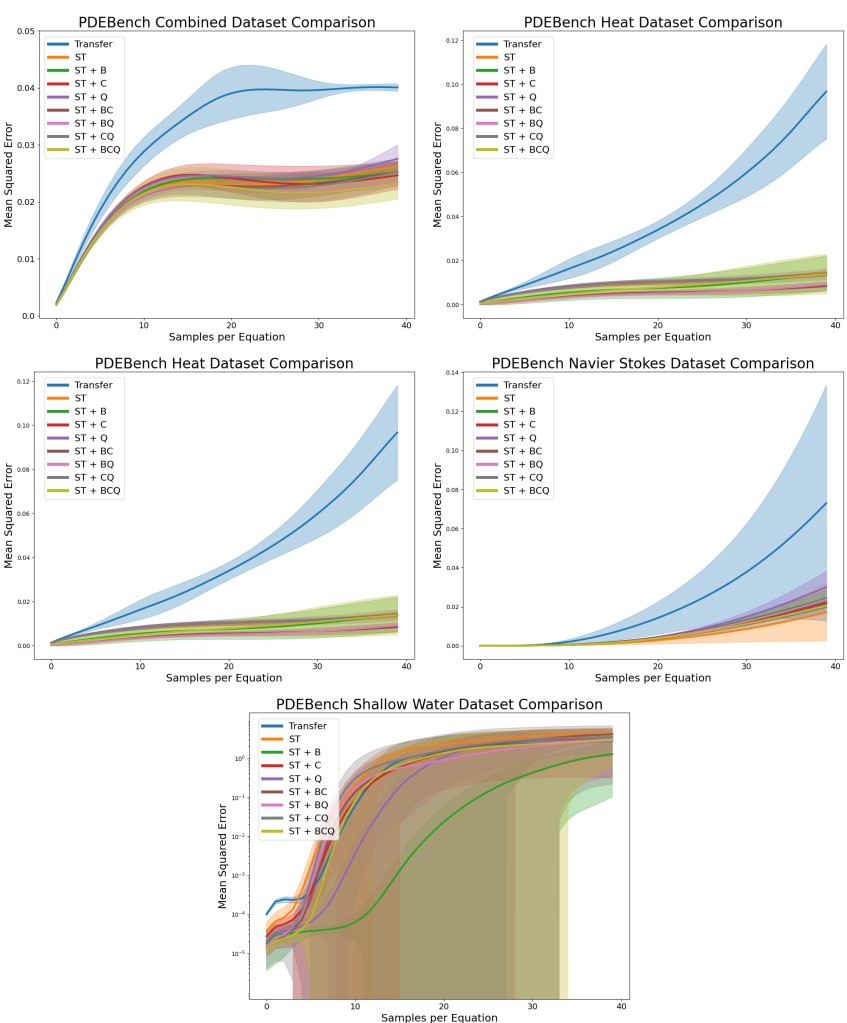

Figure 10: Autoregressive rollout MSE for each dataset.

# E DATA SCALING ADDITIONAL RESULTS

Data scaling results for the combined data sets are given in figure 11, for the Navier-Stokes data set is given in 12, and for the Shallow Water equations is given in 13.

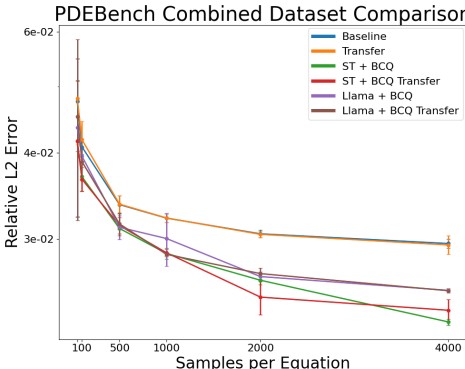

Figure 11: Comparison of next-step prediction relative $L^2$ error for baseline FactFormer, FactFormer + ST, and Factformer + Llama with and without transfer learning as the amount of data increases.

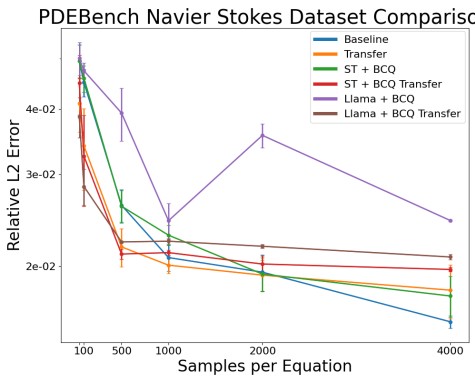

Figure 12: Comparison of next-step prediction relative $L^2$ error for baseline FactFormer, FactFormer + ST, and Factformer + Llama with and without transfer learning as the amount of data increases.

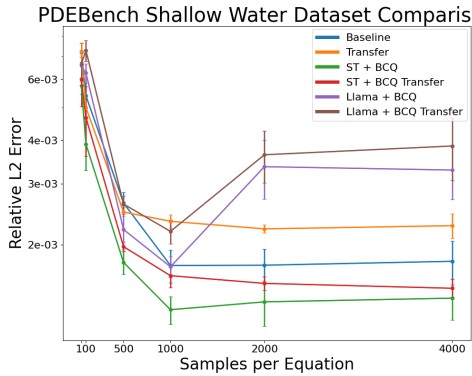

Figure 13: Comparison of next-step prediction relative $L^2$ error for baseline FactFormer, FactFormer + ST, and Factformer + Llama with and without transfer learning as the amount of data increases.

# F  T-SNE ADDITIONAL RESULTS

We see as we increase sentence complexity, we get additional structure in our t-SNE embeddings. Adding coefficient information captures the distribution of coefficients well, seen in both figure 14 and figure 15.

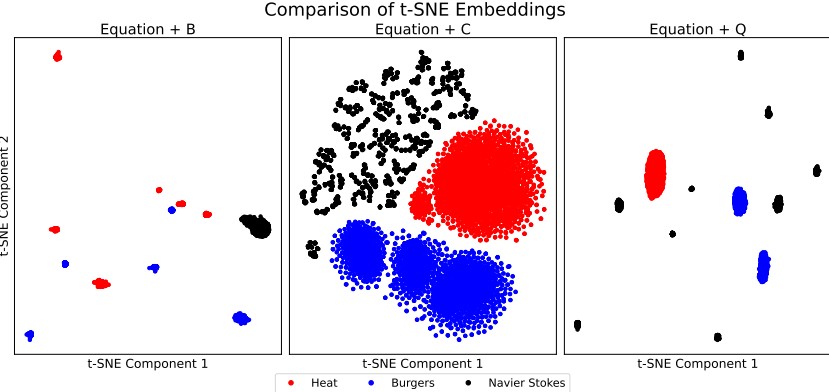

Figure 14: t-SNE embeddings for sentence-level embeddings generated by SentenceTransformer for basic equation description with each of boundary conditions, operator coefficients, and qualitative information separately.

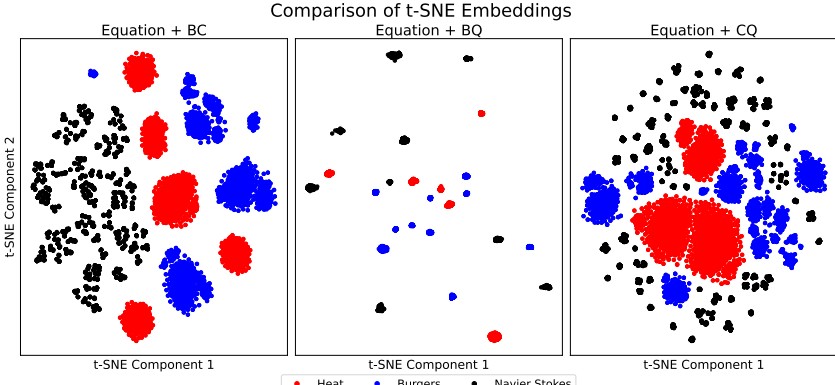

Figure 15: t-SNE embeddings for sentence-level embeddings generated by SentenceTransformer for each combination of two of boundary conditions, operator coefficients, and qualitative information.

