# OpenReview forum: "Explain Like I'm Five: Using LLMs to Improve PDE Surrogate Models with Text"
_ICLR.cc/2025/Conference — ICLR 2025 Conference Withdrawn Submission_

### Official Review · Reviewer_E1kX · 2024-10-24

**Soundness:** 2
**Presentation:** 2
**Contribution:** 2
**Rating:** 3
**Confidence:** 4

**Summary:**

This paper proposes leveraging a large language model (LLM) to incorporate multimodal information in training a machine learning (ML) surrogate model for partial differential equations (PDEs). Unlike prior studies that provided only brief or limited text descriptions of the target system, the authors argue that a more comprehensive text description enhances model performance. Their claims are partially supported by experimental results on various 2D datasets.

**Strengths:**

S1: The paper attempts to include more detailed text descriptions of the target system compared to previous work, such as PDE parameters, PDE descriptions, and boundary conditions.

S2: The effectiveness of the comprehensive text descriptions is at least partially validated by experimental results.

**Weaknesses:**

Major Concerns: The following issues should be addressed before acceptance:

W1. $\textbf{The parameter ranges used to generate data may not be adequately described for the chosen resolution.}$ The authors employ relatively small diffusion-type coefficients for the heat equation and Navier–Stokes equation, even at a low resolution of 64x64. Generally, numerical viscosity is estimated using the system's Reynolds number, and the resolution should exceed the Reynolds number to accurately capture physical viscosity or diffusion coefficients. Based on this principle, the chosen viscosity and diffusion coefficients are too small for the current resolution, meaning the generated data may reflect numerical rather than physical viscosity. The authors should provide a convergence check results for all the generated data (or at least small dissipation coefficient regime) to ensure that the generated data are resolution-independent and properly reflect the used PDE parameters.

W2. $\textbf{The analysis of the impact of the text descriptions is insufficient.}$ According to Appendix A, only one type of sentence description was explored. This makes it difficult to assess the contribution of the text description, which is a key claim of the paper. A more systematic investigation of how the detailed system descriptions affect the model's performance is needed, such as testing different sentence formats, identifying important words, and exploring how shorter or longer descriptions influence outcomes.

W3. $\textbf{A more systematic ablation study is needed.}$ Related to W2, the ablation study (Section 5.4) lacks depth in analyzing sentence descriptions. While Table 2 offers interesting information (e.g., BCQ is ineffective in most cases), the authors provide only a brief discussion (one sentence). A more thorough analysis of this part would strengthen the paper's impact and rigor.

Minor Concerns: The following are suggestions for improvement:

W4. It would be beneficial to include another backbone model. Currently, only FactFormer is used, and it is unclear whether the results are specific to this architecture. Testing with different backbones would allow for a more general analysis.

W5. FactFormer is not a conditional model and does not accept PDE parameters, which weakens the baseline comparison with the proposed model. A more convincing evaluation would include comparisons with recent models, such as UPS or a conditional model like the Neural PDE Solver (Brandstetter et al., 2022).

W6. Internal author comments (e.g., at line 165 [height] and line 444 [analyze further?]) remain in the manuscript. These should be removed before submission.

W7. In the introduction, the final sentence of the first paragraph lacks citation and the term "optimally" is unclear. Additionally, in the second paragraph, it is unclear if the phrase "these models" includes physics-informed neural networks (PINNs), which are not data-driven models. The authors should refine the English and provide appropriate citations.

**Questions:**

Q1. Figure 3: Why does BCQ information not work for the Navier–Stokes case? Does performance change with different text descriptions? It is also unclear why the more challenging autoregressive case in Table 1 does not follow the trend seen in Figure 1 (BCQ improved performance in the Navier–Stokes case in Table 1, which is difficult to reconcile with the present explanation).

Q2. Figure 6: How do the authors conclude that the embedding of "Equation + BCQ" is superior to "Equation" alone based on this figure? The distribution of different equation embeddings appears uniform, which could confuse downstream task layers.

Q3. Section 5: The definition of "transfer learning" is missing in the main body. It would be helpful to include a brief explanation of what was done for transfer learning in the main body.

Q4. Page 3, line 159 (Sec 3.1): What is meant by "2 seconds"? Does this refer to physical time or computational time unit? If the former, why is 2 seconds considered a meaningful timescale?

For additional comments, please refer to the "Weaknesses" section.

---

### Official Review · Reviewer_sKQv · 2024-10-28

**Soundness:** 1
**Presentation:** 2
**Contribution:** 1
**Rating:** 3
**Confidence:** 5

**Summary:**

This work introduces a multimodal framework for solving partial differential equations by embedding numerical data with FactFormer and encoding textual information (such as governing equations, boundary conditions, or coefficient information) through a pre-trained language model. The framework employs a cross-attention module to fuse embeddings from both modalities, facilitating the integration of system information into the numerical surrogate model. The authors provide several numerical experiments to validate the performance of the proposed method.

**Strengths:**

The paper tackles the integration of textual and numerical data within PDE surrogate models. By using a cross-attention module to combine FactFormer’s numerical embeddings with language model embeddings of system information, the framework provides a structured approach to incorporate diverse sources of information. The experiments demonstrate improvements in next-step prediction and autoregressive rollout tasks, showing the potential of multimodal frameworks for PDE applications.

**Weaknesses:**

1. The approach does not compare with existing state-of-the-art methods for solving PDEs, such as the Fourier Neural Operator and DeepONet. Additionally, numerous transformer-based methods have been developed for PDEs, but these are not referenced or compared here.
2. Multimodal models for solving PDEs already exist. For instance, the Unified PDE Solver (https://arxiv.org/abs/2403.07187) leverages both numerical and textual embeddings to solve PDEs. The lack of a direct comparison with existing models makes it challenging to assess the effectiveness of the proposed method over prior multimodal approaches.
3. The testing data includes only five relatively simple PDEs, which restricts the evaluation of the model’s capabilities. The paper would benefit from testing on more complex PDE data, such as those in the PDEBench, PDEArena, or CFDBench datasets, to better validate its performance and generalizability.
4. There are other multimodal approaches (numerical and textual data) that solve differential equations, such as FMint (https://arxiv.org/abs/2404.14688) and PROSE-FD (https://www.arxiv.org/abs/2409.09811). A discussion of such related work would enhance the impact of this paper’s contributions.

**Questions:**

Please see weaknesses.

---

### Official Review · Reviewer_5NvZ · 2024-10-31

**Soundness:** 2
**Presentation:** 1
**Contribution:** 1
**Rating:** 3
**Confidence:** 3

**Summary:**

This submission follows the line of work on neural-network-based surrogate models. The proposal is a technique for enhancing neural PDE solvers with large language models, called a "multimodal surrogate model".
Concretely, the algorithm goes as follows: the LLM is prompted with a textual description of the PDE and generates embeddings. These embeddings are combined with the data (via cross-attention) and a FactFormer architecture.

The driving question for assessing this new method is now whether or not the LLM-generated embeddings improve the reconstruction, compared to (i) a "raw" FactFormer and (ii) existing methods for combining LLMs and neural PDE solvers.

**Strengths:**

The submission's biggest strength is that the LLM embeddings consistently improve the reconstruction compared to the FactFormer without LLM augmentation. For example, Figure 3 and Table 1 demonstrate how the LLM-generated embeddings improve the reconstruction error for both next-step prediction and autoregressive rollout compared to the FactFormer.
In other words, point (i) from my summary above is answered with a strong "yes".

However, the answer to point (ii) is less clear. I discuss this under "Weaknesses" below.

**Weaknesses:**

The submission's strength is that the algorithm improves the reconstruction compared to a FactFormer. However, I consider the following weakness to be significant enough so that I recommend rejection:

Even though the idea of combining LLMs with PDE solvers has not been explored much in the literature, it is not new; for example, Shen et al. (2024) proposed something similar to the submitted manuscript.
I find that the main weakness of the submission is that it does not compare thoroughly to Shen et al.'s (2024) work on "Universal Physics Solver" (UPS).
The only mention of UPS is in Line 044 on page 1; however, I believe that a thorough discussion in Section 2 and direct comparisons in all experiments is essential to assessing the performance of the suggested algorithm.
Other related works are also not discussed prominently enough, like Zhou et al.'s (2024) Unisolver, but the lack of discussion of UPS stood out the most.
Similarly, the context of other state-of-the-art surrogate models, e.g., the Fourier Neural Operator, would be helpful for the experiments but is omitted in the submission.

In other words, the comparison to the FactFormer is relevant, but more baselines are needed. Concretely, I would like to see methods similar to those reported by Takamoto et al. (2023b) or Shen et al. (2024). Without benchmarks involving those algorithms, I cannot recommend acceptance.

To change my mind, I would have to be convinced that I misunderstood something in that UPS or Unisolver did something fundamentally different to the submission and that neural operators and other neural-network-based surrogate models are not meaningful comparisons. Another way to change my assessment would be to run these benchmarks with a competitive performance of the submitted algorithm. I suspect that re-running the benchmarks thoroughly using all these new baselines is too much work for a revision period. However, I am looking forward to possibly being proven wrong.

**Questions:**

Sometimes, I find the presentation a bit difficult to follow. For example, Section 4.2 is difficult to contextualise to prior work: how much of Section 4.2 is the FactFormer, and how much is new? The section might be easier to read if it came with a figure or an algorithm box. However, since this stylistic point is subjective (and would be relatively easy to fix if the authors agree with me), it is less relevant to my acceptance/rejection decision than the weakness outlined above.

If time and rebuttal space allow, I would also like to see the answers to the following questions.
They are less critical for my recommendation than everything written above, but I think their answers might improve the paper's clarity.

- Why are there no whitespaces before all \citep's? E.g. Lines 041f
- Figure 1: Is it on purpose that data input and data output are identical?
- Line 074: To me, it is not apparent whether more instances of PDE problems make the dataset harder or easier. What is the motivation for this statement?
- Figure 3: What does the figure look like with a logarithmic y-axis?
- Line 307: Are three random seeds sufficient? There seems to be significant variation in Figure 3.
- Line 304: Is the autoregressive rollout error MSE normalised in any way? The accumulated RMSEs are surprisingly large. Related:
- Table 1: Why is the 10^2 removed from the table values? It simply shifts the comma by two digits, and it seems that no digits are saved.
- Table 1 (again): Since the MSEs are so large, can we say anything's been learned reliably? It would be interesting to see plots of the predicted PDE solutions (for a test data point, for example, using all involved methods).
- Line 345: Where do these percentages come from? I might be missing something obvious, but I struggle to link them to the values in Tabe 1.

---

### Official Review · Reviewer_dkhU · 2024-11-04

**Soundness:** 3
**Presentation:** 3
**Contribution:** 2
**Rating:** 5
**Confidence:** 2

**Summary:**

Solving PDEs is crucial in science and engineering, yet high computational demands have driven the adoption of machine learning for faster solutions. Traditional methods lack system-specific information, but recent LLMs allow text-based data integration. This study uses pretrained LLMs to embed system information into PDE learning, outperforming the FactFormer baseline in predictive tasks across multiple equations. Analysis shows that pretrained LLMs create structured latent spaces aligned with system data.

**Strengths:**

Completed work: The article is well-written, with clear explanations that make the complex material accessible. The charts and visuals used effectively enhance the presentation, making the findings easy to follow and engaging.

Well-organized structure: The paper is structured thoughtfully, particularly in the methodology section. The clear separation of system descriptions and model details adds clarity, as does the organization of the experimental setup, which is presented in four different experiment setting clearly.

Sufficient experiments: The experiments are extensive and thorough, as sufficient support for the method.

**Weaknesses:**

Data generation graph: It would be helpful to clarify why the data generation section appears so early in the paper. Explaining the rationale behind its positioning could improve readability and guide the reader through the study’s flow.

Mathematical Support: The paper only illustrates the method part in section 4.2 which makes it hard for me to understand the whole theory behind the proposed method. More detailed mathematical backing could provide readers with a deeper understanding of the methods and strengthen the study’s credibility.

Notation formatting: There are some inconsistencies and minor errors in notation and formatting that could be polished. For example,

(1) In line 184, the notation $256x256$ is represented as character "$x$" which looks not professional. Instead, it should be $256\times256$.

(2) The text descriptions in Section 4.1 are middle, where I think left-aligned might enhance readability.

(3) In line 250 has a typo where “leaning” should be “learning.”

Experiment Labeling: In the experiment section, it would be clearer if your method were explicitly labeled with "Ours", making it easier for readers to identify and interpret the results related to your approach.

**Questions:**

In line 52, the author claims that “we aim to more fully utilize LLM understanding in PDE surrogate modeling and incorporate text information into neural operators.” suggests a comprehensive integration of LLM capabilities in PDE modeling. However, based on my understanding, the role of the LLM is limited to converting text descriptions into embeddings. How this step "fully utilized LLM"? I may ask for more illustrations about that.

In section 4, The paper gives an example of "full sentence descriptions of systems" for Burger's equation,  then what about not "full description” within this context? How would the model handle cases where only partial or incomplete information is available for an equation?

---

### Note · Authors · 2024-11-25

**Comment:**

After careful consideration, we have decided to withdraw the paper. Thank you to the reviewers who took time to carefully review our submission. The comments offer good suggestions for ways to improve this work going forward.

**Withdrawal Confirmation:**

I have read and agree with the venue's withdrawal policy on behalf of myself and my co-authors.